# The Effect of HSV-1 Seropositivity on the Course of Pregnancy, Childbirth and the Condition of Newborns

**DOI:** 10.3390/microorganisms10010176

**Published:** 2022-01-14

**Authors:** Irina Anatolyevna Andrievskaya, Irina Valentinovna Zhukovets, Inna Victorovna Dovzhikova, Nataliya Alexandrovna Ishutina, Ksenia Konstantinovna Petrova

**Affiliations:** 1Laboratory of Mechanisms of Etiopathogenesis and Recovery Processes of the Respiratory System at Non-Specific Lung Diseases, Far Eastern Scientific Center of Physiology and Pathology of Respiration, 22 Kalinina Str., 675000 Blagoveshchensk, Russia; dov_kova100@rambler.ru (I.V.D.); ishutina-na@mail.ru (N.A.I.); ksuha20031982@mail.ru (K.K.P.); 2Department of Obstetrics and Gynecology, Faculty of Postgraduate Education, Amur State Medical Academy of the Ministry of Health of Russia, 675000 Blagoveshchensk, Russia; zhukovec040875@mail.ru

**Keywords:** herpesvirus infection, pregnancy, childbirth, newborns, congenital HSV-1 infection

## Abstract

The goal of this research was to evaluate seropositivity to HSV-1 among pregnant women and its effect on the course of pregnancy, childbirth and the condition of newborns. Methods: The serological status, socio-demographic characteristics, parity of pregnancy and childbirth and condition of newborns in women seronegative and seropositive to HSV-1 with recurrent infection and its latent course during pregnancy were analyzed. Newborns from these mothers made up the corresponding groups. Results: Low titers of IgG antibodies to HSV-1 in women in the first trimester of pregnancy are associated with threatened miscarriage, anemia in pregnancy and chronic placental insufficiency. High titers of IgG antibodies to HSV-1 in women in the second trimester of pregnancy are associated with late miscarriages and premature births, anemia in pregnancy, chronic placental insufficiency, labor anomalies, early neonatal complications (cerebral ischemia, respiratory distress syndrome) and localized skin rashes. Low titers of IgG antibodies to HSV-1 in women in the third trimester of pregnancy are associated with premature birth, anemia in pregnancy, chronic placental insufficiency, endometritis, complications of the early neonatal period and localized skin rashes. Conclusions: Our research showed that low or high titers of IgG antibodies to HSV-1, determined by the timing of recurrence of infection during pregnancy, are associated with a high incidence of somatic pathology and complications in pregnancy, childbirth and the neonatal period.

## 1. Introduction

Modern trends in clinical medicine are determined by a change in the spectrum and progression of infectious diseases and an increase in the proportion of opportunistic infections, the development of which is determined by immunodeficiency. One of these infections is herpesvirus infection, caused by herpes simplex virus (HSV) types 1 and 2, which has a high level of seroprevalence among the adult population of the planet [1]. According to WHO (latest data for 2016), about 67% of the world’s adult population aged 15–49 years is infected with HSV-1 and 13% with HSV-2. Recurrent forms of the disease affect 12–25% of the adult population, where 30% of the infection occurs in subclinical and latent forms [2]. A distinction of HSV is its association with other infections. In 50–80% of women, the infection is registered as mixed: with cytomegalovirus (CMV) in 18–20% and with chlamydia in 9–21%, with opportunistic flora in 5–10% [3]. The highest incidence rate is registered among women of reproductive age from 20 to 29 years old, which determines the medical and social significance of the problem [4]. It should also be noted that immune modulation during pregnancy can also increase susceptibility to viral infections during pregnancy [5]. The spread of HSV in a population usually occurs as a result of the reactivation of a latent virus in infected neurons of the sensory ganglion and anterograde axonal transport to the innervated mucosa, followed by the replication of the virus in the epithelium and its secretion [6]. Grouped vesicular and/or ulcerative lesions typical of HSV may or may not occur during these episodes, and subclinical HSV shedding is just as common in individuals without a history of lesions as in patients with such a history [7]. The prevalence of HSV among pregnant women in the United States is 22–36%, while in Russia there are up to 14.5 cases per 100 thousand people [8]. At the same time, the total prevalence of HSV among pregnant women is 72%. According to a multicenter study, the prevalence of HSV-1 serotype among pregnant women in the United States was 63%, while the prevalence of HSV-2 was 22% [9]. Despite numerous studies, the true prevalence of HSV and the level of seropositivity among pregnant women worldwide remains unclear.

Intrauterine herpes infection in the first and second trimester of pregnancy leads to adverse outcomes of spontaneous miscarriage and fetal malformations [10], while in the third trimester, it can lead to premature birth, antenatal hypotrophy, hypoxia and fetal death [11]. One of the possible causes of pregnancy complications in HSV is the infection of the placenta, which can cause chorio-decidual changes affecting fetal development [12]. It has been shown that in 2.2–8.4% of asymptomatic pregnant women with normal ultrasound results, viral genomes are found in the amniotic fluid [13,14], which increases the risk of intrauterine infection. The frequency of infection of a newborn in the presence of a primary herpes infection in a pregnant woman reaches 40–50%, with a recurrence rate of up to 5% [15]. Although HSV is a common infection, it is unclear why intrauterine infection is rare. Perhaps the reasons are the high circulation of maternal antibodies and the presence of effective defense mechanisms on the part of the uteroplacental region, which block the further spread of HSV [16]. At the same time, up to 85% of neonatal HSV infection is transmitted through the parturient canal [17].

Neonatal HSV infection is divisible into three clinical syndromes: localized cutaneous, eye and mouth; combined or not localized lesion of the skin, eyes, mouth and central nervous system (CNS); and disseminated disease that includes a spread to internal organs. The advanced disease may or may not affect the central nervous system and can lead to hepatitis, pneumonitis, sepsis, multiple organ failure syndrome and disseminated intravascular coagulation. The disease appearing with cutaneous and mucosal infection accounts for approximately 45% of neonatal HSV cases, although these infants often develop central CNS injury or disseminated disease if untreated, accounting for approximately 30% and 25% of cases, respectively. Both HSV-1 and HSV-2 can cause localized skin and mucosal lesions, CNS or disseminated disease, although CNS infection with HSV-2 is associated with greater morbidity [18]. Newborns with severe HSV infection often do not have clinical symptoms, which often leads to adverse consequences when the diagnosis is made and the initiation of antiviral therapy is delayed [19]. Based on the above, the identification of women with a high risk of HSV transmission to newborns on the basis of demographic or clinical characteristics and their timely serological monitoring is a potentially justified strategy for creating measures for the prevention and treatment of neonatal infection. The public health benefit of this approach is clear.

The present research is aimed at assessing seropositivity for HSV-1 among pregnant women and its effects on pregnancy, childbirth and the condition of the newborn.

## 2. Materials and Methods

### 2.1. Research Settings and Design

A retrospective analysis of medical documentation (individual cards of a pregnant and postpartum woman (form No. 111/y), medical records of an inpatient (form No. 003/y), birth history (form No. 096/y)) of 150 pregnant women and postpartum women and their newborns for the period from January 2018 to March 2020, registered in the antenatal clinic No. 2 of Amur region autonomous public health care institution “Blagoveshchensk City Clinical Hospital”, was performed.

Serological status, socio-demographic characteristics, the ratio of pregnancies and childbirth and the condition of newborns were analyzed in 150 pregnant women.

All pregnant women were comparable in terms of their average ages, which in seropositive women with recurrent HSV-1, seropositive women without recurrent HSV-1 and women seronegative for HSV-1 and HSV-2 were 25.9 ± 0.4, 27.4 ± 0.8 and 26.9 ± 0.8 years (*p* > 0.05), respectively. The pregnancy outcomes in terms of the number of births and legal abortion were comparable in the studied groups of women (*p* > 0.05). All women were residents of the city.

Inclusion criteria for the study were as follows: (1) HSV-1 seropositive pregnant women; (2) residents of Blagoveshchensk; (3) age 18 years and older; (4) consent to participate in the research. HSV-2 seropositivity in pregnant women was an exclusion criterion.

### 2.2. Ethical Review

The research was carried out in accordance with the Order of the Ministry of Health of the Russian Federation of 1 November 2012 No. 572n (as amended on 17 January 2014) “Concerning Approval of the Procedure for the provision of medical care in the profile of” obstetrics and gynecology (except for the use of assisted reproductive technologies) “and with the approval of the Committee on biomedical ethics of the Far Eastern Scientific Center of Physiology and Pathology of Respiration” (Blagoveshchensk, Russia). Written consent for the collection and processing of various data was obtained from all participants in the study.

### 2.3. Serological Tests of HSV

Serum samples were tested for the presence of nonspecific HSV-1,2 (Vecto-HSV-1,2-IgM and Vecto-HSV-1,2-IgG, VectorBEST, Novosibirsk, Russia) and specific immunoglobulins (Ig) HSV-2 class M and G (“Vecto-HSV-2-IgM” and “Vecto-HSV-2-IgG”, VectorBEST, Russia) in accordance with the instructions for their use. The main reagents of the kit for detecting IgG HSV-1, 2 are an immobilized HSV antigen based on fragments of glycoproteins (g) G and gC; for HSV-2, a recombinant protein with a corresponding fragment of gG was used as an antigen.

The sensitivity and specificity of the kits for HSV-1, 2 and HSV-2, according to the manufacturer’s recommendations, is 100% (the interval is 97.8–100% with a confidence level of 90%).

Based on the data obtained, the critical value of optical density (ODcrit) was calculated by the formula ODcrit = ODav K^−^ + 0.20 (ODav K^−^ is the arithmetic mean value of optical density in the wells with a negative control sample).

The result of the analysis was considered as positive if the optical density of the sample was greater than or equal to the critical values of the optical density of the negative control. The result of the analysis was considered negative if the optical density of the sample was less than the critical values of the optical density of the negative control. The determination of the titer of IgG antibodies to HSV-1, 2 in positive samples was carried out by titration in accordance with the manufacturer’s instructions. The titer was the last dilution of the test serum at which the optical density of the sample in the corresponding well exceeded the average optical density of the negative control by 0.1 relative unit.

### 2.4. Data Analysis

Data entry and analysis were performed using the IBM SPSS Statistics application package version 23.0 (Armonk, NY, USA). The age of the research participants and the onset of menarche and the weight of the body weight of newborns in independent groups with a normal distribution of random variables and a fixed variance were determined using the unpaired parametric Student’s *t*-test. Data are presented as arithmetic mean and standard deviation (M ± SD). The comparison of the frequencies of the alternative distribution of socio-demographic and clinical signs (parity of pregnancy and childbirth, the state of newborns) was carried out using the Fisher test. Statistically significant differences were defined as *p* < 0.05. All variables with a *p* value < 0.05 were included in the regression models and were presented as relative risks (RR) with their 95% confidence intervals (CI) (low and high).

## 3. Results

### 3.1. Dynamics of HSV-1 Seropositivity among Pregnant Women

In total, we examined 150 serum samples, of which 30 were negative for nonspecific IgM and G to HSV-1,2 and specific IgM and G to HSV-2. In total, 120 samples were positive for non-specific IgM and G to HSV-1, 2 and negative for specific IgM and G to HSV-2. A negative serological reaction to HSV-2 suggested that these study participants were latent carriers of HSV-1 only. Positive samples were titrated over time after 12 days. In 90 study participants, there was a fourfold or greater increase in IgG antibody titers, as well as the absence (negative test) of IgM and G to HSV-2, which can serve as a criterion for the current disease. The absence of IgM and G to HSV-2 and the absence of an increase in the level of IgG antibodies to HSV-1, 2 is considered as an asymptomatic, non-recurrent form of HSV-1 infection. Titers of antibodies IgG 1:12,800 and 1:6400 were regarded by us as high, 1:3200 and 1:1600 as moderate and 1:800 as low.

Indicators of IgG antibody titers in the group with recurrent HSV-1 and in the group with no recurrence of HSV-1 in the first trimester of pregnancy are presented in Table 1. High titers of IgG antibodies were detected in 36.7% of participants in the group with recurrent HSV-1, which is higher (*p* < 0.05) than in the group with no recurrence of HSV-1 (3.3%). An IgG antibody titer 1:12,800 was detected in 16.7% of participants in the group with recurrent HSV-1, and an antibody titer 1:6400 in 20% of participants (*p* < 0.05), which is 1.33 times (1.13–1.56) more often than in the group with no recurrence of HSV-1 (3.3%). Moderate values of IgG antibody titers 1:3200 and 1:1600 were found in 18.9% and 23.3% of participants in the group with recurrent HSV-1, which is significantly lower (*p* < 0.05) than in the group with no recurrence of HSV-1 (36.7% and 46.7%, respectively). The frequency of detecting low titers of IgG antibodies 1:800 in the group with recurrent HSV-1 and in the group with no recurrence of HSV-1 was not statistically significant (21.1% and 13.3%, respectively).

Thus, the recurrence of HSV-1 in the first trimester was often associated with the detection of high (36.7%) and moderate (42.2%) IgG antibody titers. No recurrence of HSV-1correlated with the detection of moderate IgG antibody titers (83.4%).

In the group with recurrent HSV-1 in the second trimester of pregnancy, the number of participants with moderate (44.4%) (*p* < 0.05) and low (45.6%) IgG antibody titers increased (Table 2). High titers of IgG antibodies were detected in 10% of participants in the group with recurrent HSV-1, of which 4.4% had IgG antibody titers 1:12,800 and 5.6% had IgG antibodies titers 1:6400 (*p* < 0.05), which is significantly lower than in the group with no recurrence of HSV-1 (33.3%). Moderate titers of IgG antibodies in the group with HSV-1 recurrence in the second trimester were detected in 44.4% of participants, of which IgG antibody titers 1:3200 were found in 8.8% of participants (*p* < 0.05), IgG antibody titers 1:1600 were found in 35.6% of participants (*p* < 0.05). The antibody titer 1:1600 was found 1.34 times (1.12–1.60) more often than in the group with no recurrence of HSV-1. In the group with no recurrence of HSV-1, IgG antibody titers 1:3200 were detected in 56.7% of participants, and IgG antibody titers 1:1600 were found in 10% of participants. Low titers of IgG antibodies 1:800 were detected in 45.6% of participants in the group with HSV-1 recurrence in the second trimester of pregnancy.

Consequently, the recurrence of HSV-1 in the second trimester of pregnancy was often associated with moderate (44.4%) and low (45.6%) IgG antibody titers. No recurrence of HSV-1 correlated with moderate IgG antibody titers (66.7%).

In the group with recurrent HSV-1 in the third trimester of pregnancy, high titers of IgG antibodies were detected in 26.6% of the participants, which is not significant (*p* > 0.05) compared with the group with no recurrence of HSV-1 (40%) (Table 3). Antibody titers IgG 1:12,800 were detected in 3.3% of participants in the group with recurrent HSV-1. The detection rate of IgG antibody titers 1:6400 in the group with recurrent HSV-1 and in the group with no recurrence of HSV-1 did not differ significantly (*p* > 0.05) and amounted to 23.3% and 40%, respectively. In 66.7% of the participants in the group with HSV-1 recurrence in the third trimester of pregnancy, moderate antibody titers were revealed: in 36.7% of participants, IgG antibodies titers 1:3200 were found (*p* < 0.05), which is lower than in the group with no recurrence of HSV-1 (50%); 30% had IgG antibody titers 1:1600 (*p* < 0.05), which is higher than in the group with no recurrence of HSV-1 (6.7%). Antibody titers IgG 1:1600 were found 1.35 times (1.14–1.59) more often than in the group with no recurrence of HSV-1. Low titers of IgG antibodies 1:800 were detected in 6.7% of participants in the group with recurrent HSV-1 in the third trimester of pregnancy and in 3.3% of participants in the group with no recurrence of HSV-1.

Consequently, the recurrence (66.7%) and no recurrence of HSV-1 (56.7%) in the third trimester of pregnancy were associated with moderate IgG antibody titers.

The comparison of serological data in the group with recurrent HSV-1 showed (Table 4) that high titers of IgG antibodies are 1.9 times more common in the first trimester than in the second trimester. Moderate titers of IgG antibodies are more common in the third trimester than in the first and second trimester by 1.67 times and 1.6 times, respectively. Low titers of IgG antibodies are more common in the second trimester than in the first and third trimesters by 1.67 times and 2.37 times, respectively.

In the group without recurrent HSV-1, high titers of IgG antibodies are 2.23 times more common in the second trimester and 2.41 times more common in the third trimester than in the first trimester. Moderate titers of IgG antibodies are 2.14 times more common in the first trimester than in the third trimester. Low titers of IgG antibodies occur 1.69 times more often in the first trimester than in the third trimester.

The results obtained reflect the susceptibility to HSV-1 in women at different stages of pregnancy, which can affect the course and outcome of pregnancy and childbirth and the condition of newborns.

### 3.2. Socio-Demographic Characteristics and Pregnancy Parity in Women with HSV-1

In terms of social status, 83.3% of the participants in the group with HSV-1 recurrence during pregnancy were employees, 12.2% were housewives and 4.4% were students; in the group with no recurrence of HSV-1, these values were 90%, 6.7% and 3.3%, respectively; and in the group of seronegative women, these values were 86.7%, 6.7% and 6.7%, respectively (Table 5). The frequency of clinical manifestations of HSV-1 during the year before the onset of present pregnancy in the group with HSV-1 recurrence during pregnancy was 5–6 times, while the frequency in the group with latent infection was 1–2 times. Clinically, the disease was manifested by general weakness; increased body temperature; rashes on the mucous membrane of the lips, the skin of the nasolabial triangle and the wings of the nose; and enlargement of the submandibular nodes. The duration of disease recurrence was on average 3–5 days.

Among the participants in the group with HSV-1 recurrence during pregnancy, 10% planned a present pregnancy, which is significantly lower than in the group with no recurrence of HSV-1 and in the group of seronegative women (*p* < 0.05, respectively).

At the onset of pregnancy, all study participants registered with the antenatal clinic for up to 12 weeks of gestation. The first clinical manifestation of HSV-1 in the first trimester of pregnancy was noted by all study participants, while relapses in the second trimester were found in 25 participants (27.8%) and relapses in the third trimester were found in 32 participants (35.6%). The recurrent course of the disease throughout pregnancy was observed in 10 (11.1%) participants.

Among the participants in the studied groups, the number of primiparous and re-pregnant but primiparous respondents did not differ significantly (*p* > 0.05). Among multiparous respondents, there were significantly more participants (27.8%) in the group with HSV-1 recurrence during pregnancy (*p* < 0.05) than in the group with no recurrence of HSV-1 (10%) and in the group of seronegative women (16.7%). Of 25 multiparous women in the group with HSV-1 recurrence during pregnancy, two (8%) noted the death of children in infancy. In both cases, the cause was intrauterine infection with HSV-1 in a generalized form. In 20 (22.2%) re-pregnant but primiparous participants in the group with HSV-1 recurrence during pregnancy, there were 1–2 spontaneous abortions in the anamnesis, which is more frequent than in the group of seronegative women (*p* < 0.05). Legal abortions were noted by participants in all groups equally often (*p* > 0.05). The participants of the group with HSV-1 recurrence during pregnancy underwent the termination of pregnancy for medical reasons by the fetus due to malformations incompatible with life. In total, 4.4% of the group members had a missed abortion.

All group participants reported maternal and paternal hereditary burdens with the same frequency (*p* > 0.05). Matrilinear inheritance in the study groups was tainted by hypertensive disease, oncopathology and diabetes mellitus with the same frequency (*p* > 0.05). Congenital heart disease was indicated equally often by participants in the group with HSV-1 recurrence during pregnancy and in the group with no recurrence of HSV-1 (*p* > 0.05). Patrilineal inheritance in the studied groups was tainted by hypertensive disease and oncopathology with the same frequency (*p* > 0.05). Diabetes mellitus was equally often indicated by the participants in the group with HSV-1 recurrence during pregnancy and in the group with no recurrence of HSV-1 (*p* > 0.05). Congenital kidney disease was reported by 1.1% in the group with HSV-1 recurrence 1 during pregnancy.

Somatic diseases were detected in 57.8% of the participants in the group with recurrent HSV-1 during pregnancy (*p* < 0.05), which is 1.46 times (1.18–1.82) more often than in the group with no recurrence of HSV-1 and 1.67 times (1.35–2.07) more often than in the group of seronegative women (Table 6). Chronic pyelonephritis occurred 1.36 times (1.16–1.59) more often than in the group with no recurrence of HSV-1. The frequency of detection of chronic forms of bronchitis (13.3%), sinusitis (5.6%), tonsillitis (11.1%) and pharyngitis (4.4%) did not differ statistically significantly in the study groups (*p* > 0.05). Of all the participants in the group with HSV-1 recurrence during pregnancy, two or more pathologies of respiratory diseases were detected in 8.9% of participants.

Acute respiratory infections (ARI) were detected in 65.6% of participants in the group with HSV-1 recurrence during pregnancy, which is 1.67 times (1.20–2.31) more often than in the group with no recurrence of HSV-1 and 2.24 (1.59–3.17) more often than in the group of seronegative pregnant women. In every third case, there was an acute onset of the disease with a rise in body temperature to 38 °C for 3 days. In 7.8% of cases, ARI was complicated by acute sinusitis, while in 2.2% of cases, it was complicated by the exacerbation of chronic bronchitis. Acute bronchitis was noted equally often (3.3%) by participants in the group with HSV-1 recurrence during pregnancy and with no recurrence of HSV-1 (*p* > 0.05).

Gynecological diseases were detected in 71.1% of participants in the group with HSV-1 recurrence during pregnancy, which is 1.59 times (1.21–2.08) more often than in the group with no recurrence of HSV-1 and 1.95 times (1.47–2.57) more often than in the group of seronegative women. The predominant diseases in the group with HSV-1 recurrence during pregnancy were cervical erosion and ectropion (29%) and chronic salpingo-oophoritis (21.1%) (*p* < 0.05). Cervical erosion and ectropion occurred 1.22 times (1.01–1.48) more often than in the group with no recurrence of HSV-1 and 1.34 times (1.13–1.58) than in the group of seronegative women. Salpingo-oophoritis occurred 1.34 times (1.14–1.57) more often than in the group with no recurrence of HSV-1. Acute vaginitis was diagnosed in 9% of participants in the group with HSV-1 recurrence during pregnancy. The frequency of chronic inflammatory disease of the uterus and oligomenorrhea detection did not differ significantly in the study groups (*p* > 0.05).

Complications of pregnancy in the group with HSV-1 recurrence during pregnancy were 297.6 per 100 cases, while in the group with no recurrence of HSV-1 and seronegative women, these values were 73.3 and 46.7, respectively (*p* < 0.05) (Table 7).

Threatened abortion in the first trimester of pregnancy was found in 51.1% of participants in the group with HSV-1 recurrence during pregnancy, which is 1.28 times (1.04–1.57) more often than in the group of seronegative women. Threatened preterm labor was found in 21.1% of participants, which is 1.34 times (1.14–1.57) more often than in the group with no recurrence of HSV-1 and seronegative women, respectively. In total, 73.3% of participants had anemia, which is 1.83 times (1.37–2.45) more often than in the group with no recurrence of HSV-1 and 1.96 times (1.46–2.64) more often than in the group of seronegative women. All participants in the group with HSV-1 recurrence during pregnancy had clinical symptoms of threatened abortion; in 9 (10%) participants, choroidal detachment with the formation of a retrochorial hematoma was diagnosed by ultrasound, and 5 (5.6%) participants exhibited chorion presentation. Chronic placental insufficiency was diagnosed in 72.2% of participants in the group with HSV-1 recurrence during pregnancy (*p* < 0.05), which is 1.99 times (1.49–2.65) more often than in the group with no recurrence of HSV-1 and 2.13 times (1.59–2.84) more often than in the group of seronegative women. The compensated form of placental insufficiency was diagnosed in 84% of the participants (*p* < 0.05), which is 1.57 times (1.25–1.97) more often than in the group with no recurrence of HSV-1 and 1.63 times (1.30–2.04) more often than in the group of seronegative women. The detection rate of the subcompensated form of chronic placental insufficiency did not differ significantly in the group with HSV-1 recurrence during pregnancy (13.8%) and in the group with no recurrence of HSV-1 (6.7%) (*p* > 0.05). The decompensated form was found in 1.1% of the group members with HSV-1 recurrence during pregnancy. Chronic intrauterine fetal hypoxia was detected in 44.4% of participants in the group with HSV-1 recurrence during pregnancy (*p* < 0.05), which is 1.49 times (1.24–1.78) more often than in the group with no recurrence of HSV-1. The incidence of fetal growth restriction did not differ significantly in the group with HSV-1 recurrence during pregnancy (7.7%) and in the group with no recurrence of HSV-1 (3.3%) (*p* > 0.05).

### 3.3. The Course of Labor and the Early Postpartum Period in Women with HSV-1

All participants in the group with no recurrence of HSV-1 and seronegative women delivered at term per vias naturales; in the group with HSV-1 recurrence during pregnancy, 6 (6.9%) participants delivered prematurely at 36 weeks. The onset of preterm labor was spontaneous after the waters broke. The total duration of labor in the group with HSV-1 recurrence during pregnancy was significantly higher than in the group with no recurrence of HSV-1 (7.42 ± 0.33 h) and in the group of seronegative women (7.37 ± 0.48 h), at 8.90 ± 0.28 h (*p* < 0.05).

Of the 84 participants in the group with HSV-1 recurrence during pregnancy with full-term pregnancy, vaginal delivery occurred in 65 (72.2%) of cases, in which 19 children (21.1%) were delivered by caesarean section. Indications for elective cesarean section were as follows: incompetent postoperative uterine scar in multiparous women in combination with obstetric pathology, found in 6 (31.6%) participants; pelvic presentation and a big fetus weighing more than 3600 g in primiparous women, seen in 5 cases (26.3%). In the group of seronegative women, 26 (86.7%) out of 30 participants delivered per vias naturales, while in the group with no recurrence of HSV-1, this number was 25 (83.3%) (*p* > 0.05).

Abnormalities of forces of labor occurred in 46.8% of participants in the group with HSV-1 recurrence during pregnancy (*p* < 0.05) (Table 8), which is 1.35 times (1.12–1.62) more often than in the group with no recurrence of HSV-1 and 1.45 times (1.22–1.73) more often than in the group of seronegative women.

Abnormalities of forces of labor were revealed in 28% of the participants in the group with HSV-1 recurrence during pregnancy (*p* < 0.05), which is 1.31 times (1.14–1.52) more often than in the group of seronegative women. There were no significant differences between the study groups in the frequency of premature rupture of membrane (*p* > 0.05). Intrapartum and postpartum hemorrhage in participants of the group with HSV-1 recurrence during pregnancy was detected in 5.1% of cases, while placenta accrete was found in 7.7% of cases. Chronic inflammatory disease of the uterus was equally often diagnosed in the group with recurrent HSV-1 during pregnancy (19%) and in the group with no recurrence of HSV-1 (11.5%).

### 3.4. The Condition of Newborns in the Early Neonatal Period in Women with HSV-1

Sixty full-term babies were born in the group with no recurrence of HSV-1 and in the group of seronegative women. In the group with HSV-1 recurrence, 91 live babies were born during pregnancy, of which 84 (92.3%) were full-term and 7 (7.6%) were premature, and there was 1 case of dichorionic diamniotic twins. One child from the dichorionic diamniotic twins died in the early neonatal period due to sepsis.

In the group of seronegative women, all children at birth were rated 7–10 points on the Apgar scale (Table 9).

In the group with recurrent HSV-1 infection during pregnancy, 83.5% of children were born with a score of 7–10 points, 15.4% were in a state of moderate asphyxia, and one premature twin (1.1%) was in a state of high severity. The average Apgar score at 1 min in the group with recurrent HSV-1 infection during pregnancy was not statistically significant and amounted to 7.31 ± 0.13 points; in the group with no recurrence of HSV-1, the score was 7.43 ± 0.14 points, and in the group of seronegative women, this score was 7.57 ± 0.10 points. At 5 min, the average Apgar scores in the group with HSV-1 recurrence during pregnancy were 8.11 ± 0.12 points, while those in the group with no recurrence of HSV-1 were 8.12 ± 0.11 points and in the group of seronegative women were 8.50 ± 0.10 points. In the group with HSV-1 recurrence during pregnancy, the weight of two premature infants was less than 2000 g, with a weight of 2000–2499 g for three children, of which two were premature and one was small-for-gestational-age. For the rest of the indicators, no significant differences were found in the study groups (*p* > 0.05). In newborns in the group with HSV-1 recurrence during pregnancy, the incidence of diseases in the early neonatal period was 123.3 per 100 cases. In terms of the structure of diseases, infections specific for the perinatal period prevailed (*p* < 0.05) (Table 10) and were detected 1.32 times (1.12–1.54) more often than in the group with no recurrence of HSV-1. Of 91 newborns in the group with HSV-1 recurrence during pregnancy, 40.6% developed an infection in the early neonatal period (*p* < 0.05); 3.3% of newborns were diagnosed with generalized forms (meningitis, pneumonia, sepsis), skin vesicles prevailed in the structure of local forms in 21% of cases, with pyoderma in 3.3% of cases. Of the 30 newborns in the group with no recurrence of HSV-1, infections in the early neonatal period amounted to 10% in the form of skin vesicles (6.7%) and conjunctivitis (3.3%).

Cerebral ischemia in newborns in the group with HSV-1 recurrence during pregnancy was diagnosed in 26.4% (*p* < 0.05), which is 1.3 times (1.10–1.55) more often than in the group with no recurrence of HSV-1. Respiratory distress syndrome was diagnosed in 19.8% of newborns (*p* < 0.05), which is 1.32 times (1.13–1.56) more often than in the group with no recurrence of HSV-1. Neonatal jaundice was diagnosed in 6.6% in the group with HSV-1 recurrence during pregnancy.

### 3.5. Influence of HSV-1 Seropositivity in Women during Pregnancy on the Development of Complications of Pregnancy, Childbirth and the Early Postpartum Period and the Condition of Newborns

The analysis of the effect of seropositivity on HSV-1 on the development of pregnancy complications showed that, in the group with recurrent infection in the first trimester with low titers of IgG antibodies 1:800, the risk of miscarriage increased by 1.85 times (Table 10). The risk of pregnant women developing anemia with high titers of IgG antibodies 1:12,800–1:6400 in the first trimester of pregnancy was 2.25 times greater, that with moderate titers of IgG antibodies 1:3200–1:1600 was 2.10 times greater, and that with low titers of IgG antibodies 1:800 was 3.07 times greater. The risk of chronic placental insufficiency with high antibody titers was 2.18 times greater, that with moderate antibody titers was 1.94 times greater, and that with low IgG antibody titers 1:800 was 2.97 times greater. The incidence of chronic intrauterine fetal hypoxia was increased with low antibody titers to 1.64 times. High antibody titers in pregnant women with HSV-1 recurrence in the second trimester of pregnancy increased the risk of miscarriage to 2.38 times and preterm birth to 1.45 times. There was a risk of anemia, which was increased 3.85 times with high antibody titers, 1.92 times with moderate antibody titers and 1.88 times with low antibody titers. The risk of chronic placental insufficiency developing with high antibody titers was increased 3.72 times, with moderate antibody titers 1.86 times and with low antibody titers 1.82 times. The risk of chronic intrauterine fetal hypoxia was associated with high antibody levels and amounted to an increase of 1.92 times. Low antibody titers in pregnant women with HSV-1 recurrence in the third trimester of pregnancy increased the risk of threatened premature birth by 1.66 times. The risk of anemia in pregnant women in the third trimester of pregnancy with high antibody titers was increased 2.75 times, with moderate antibody titers 2.01 times and with low antibody titers 4.13 times. The risk of chronic placental insufficiency developing with high antibody titers was increased 2.66 times, with moderate antibody titers 1.94 times and with low antibody titers 3.99 times. An increased risk of chronic intrauterine fetal hypoxia was associated with the presence of high titers by 1.45 times and low antibody titers by 2.33 times in women in the third trimester of pregnancy.

The risk of birth and early postpartum period complications—namely, labor abnormalities—was associated with high antibody titers in women with HSV-1 recurrence in the second trimester of pregnancy, increasing by1.56 times, while the development of inflammatory diseases of the uterus was associated with low antibody titers in women with HSV-1 recurrence in the third trimester of pregnancy, increasing by 1.59 times (Table 11).

Complications of the early neonatal period—namely, an increased risk of cerebral ischemia by 1.45 and 1.82 times and respiratory distress syndrome by 1.45 and 1.63 times for newborns—were associated with high antibody titers in women with HSV-1 recurrence in the second trimester of pregnancy and low antibody titers with disease recurrence in the third trimester of pregnancy, respectively (Table 12). In the structure of local congenital forms of infection, the frequency of detection of skin vesicles (vesiculosis) in newborns from women with HSV-1 recurrence in the second trimester and high antibody titers increased 1.45 times, while women with low antibody titers increased 1.67 times with HSV-1 recurrence in the third trimester of pregnancy.

According to the results obtained, it can be concluded that the risk of complications of pregnancy, childbirth and the early postpartum period, congenital infection and increased morbidity in newborns is associated with low or high IgG antibodies to HSV-1, determined by the timing of infection recurrence during pregnancy.

## 4. Discussion

According to the latest data (2016–2019), overall serological prevalence rates in different countries vary from 54.5–96.4% for HSV-1 and 16.8–26.9% for HSV-2 [20,21,22]. Despite the fact that considerable attention is paid to the increase in the incidence of neonatal HSV-2 infection due to a more severe prognosis for newborns [23,24], up to 30% of HSV infections occur with HSV-1, which also suggests the social significance of the problem [25]. Both primary and recurrent maternal infection can lead to congenital disease, even if the risk of re-infection is low. There is evidence of a high risk of intrauterine infection with recurrent HSV during the first 20 weeks of pregnancy, which leads to abortion, stillbirth and congenital anomalies in surviving infants [26]. Perinatal mortality in generalized forms of HSV infection is 50% [27]. According to some data, prolonged rupture to the delivery interval [28] and invasive obstetric procedures [29] can be markers of neonatal infection. At the same time, the relationship between HSV-1 seropositivity during infection recurrence during pregnancy and the development of congenital and maternal infections associated with severe complications of pregnancy, childbirth and the early postpartum period remains unconfirmed [30,31].

We studied the change in seropositivity to HSV-1 in pregnant women at different stages of pregnancy and with different courses of infection. The comparison of serological data in the group with recurrent HSV-1 showed that high titers of IgG antibodies (1:12,800–1:6400) are much more common in the first trimester than in the second trimester. Moderate IgG antibody titers (1:3200–1:1600) are more common in the third trimester than in the first and second trimesters, respectively. Low titers of IgG antibodies (1:800) are more common in the second trimester than in the first and third trimesters, respectively. In the group without recurrent HSV-1, high titers of IgG antibodies are more common in the second and third trimester. Moderate IgG antibody titers are more common in the first trimester than in the third trimester. Low titers of IgG antibodies are more common in the first trimester than in the third trimester.

Our results reflect the susceptibility to HSV-1 in women at different stages of pregnancy, which is consistent with studies showing a high serological prevalence of HSV-1 among pregnant women in the first and third trimester of pregnancy with a high risk of recurrence of infection [32]. This suggests that these women are at greater risk of developing complications of pregnancy and transmitting HSV-1 infection to their newborn.

Further analysis showed no differences in socio-demographic characteristics among the study participants, which is confirmed by other studies [33]. At the same time, the frequency of detection of somatic (chronic pyelonephritis, acute respiratory infections, complicated by exacerbation of chronic sinusitis and chronic bronchitis) and gynecological (ectopia of the cervix, chronic salpingo-oophoritis) pathology in women with recurrent HSV-1 during pregnancy was significantly higher than without recurrence of HSV-1.

Moreover, our study found a relationship between the titer sof IgG antibodies to HSV-1 in women with recurrence of the disease and obstetric complications. It was shown that low titers of IgG antibodies to HSV-1 in women in the first trimester of pregnancy were associated with threatened miscarriage, anemia in pregnancy and chronic placental insufficiency. High titers of IgG antibodies to HSV-1 in women in the second trimester of pregnancy are associated with the threatened late miscarriages and premature birth, anemia in pregnancy and chronic placental insufficiency. Low titers of IgG antibodies to HSV-1 in women in the third trimester of pregnancy are associated with preterm birth, anemia in pregnancy and chronic placental insufficiency.

The risk of complications in the act of delivery and the early postpartum period— namely, congenital anomalies—increased in women with high titers of IgG antibodies to HSV-1 with a relapse of the disease in the second trimester of pregnancy. The development of postpartum endometritis was associated with low titers of IgG antibodies to HSV-1 in women with a relapse of the disease in the third trimester of pregnancy.

Complications of the early neonatal period—namely, the risk of cerebral ischemia in newborns and the development of respiratory distress syndrome—were associated with high titers of IgG antibodies to HSV-1 in women with relapses of the disease in the second trimester of pregnancy and low titers of IgG antibodies to HSV-1 in relapses of diseases in the third trimester of pregnancy. In terms of the structure of localized infections, the frequency of detection of skin vesicles increased in newborns from women with high titers of IgG antibodies to HSV-1 with relapses of the disease in the second trimester and low titers of IgG antibodies to HSV-1 with relapses of the disease in the third trimester of pregnancy. Our data do not contradict studies that show the role of recurrent maternal HSV infection in the development of congenital local infection in newborns [34]. Diagnosis is based on the following symptoms: skin vesicles, fever, lethargy, convulsions, conjunctivitis, pneumonia, disseminated intravascular coagulation, central nervous system damage and respiratory distress syndrome. Symptoms may be present at birth, but in most cases, they occur later than 5 days after birth, and sometimes even after 4–6 weeks of life, which leads to undesirable consequences for the newborn [35].

Our study confirmed that low or high titers of IgG antibodies to HSV-1, determined by the recurrence of infection during pregnancy, are associated with a high incidence of somatic pathology and complications of pregnancy, childbirth and the neonatal period. The results obtained are of practical importance and can be used in the formation of risk groups for obstetric complications and to prevent unfavorable perinatal outcomes in women with HSV-1 during pregnancy.

## Figures and Tables

**Table 1 microorganisms-10-00176-t001:** Frequency of distribution (%) of IgG antibodies to HSV-1 among pregnant women in the first trimester of pregnancy.

Groups	Titers of IgG Antibodies									
	High				Moderate				Low	
	1:12,800		1:6400		1:3200		1:1600		1:800	
	Abs	%	Abs	%	Abs	%	Abs	%	Abs	%
HSV-1 recurrence	15	16.7	18	20 *	17	18.9 *	21	23.3 *	19	21.1
**Total**			33	36.7 *			38	42.2 *	19	21.1
No recurrence of HSV-1			1	3.3	11	3.7	14	46.7	4	13.3
**Total**			1	3.3			25	83.4	4	13.3

Note: * *p* < 0.05 significance of differences between pregnant women with recurrent HSV-1 and in the group with no recurrence of HSV-1 in the first trimester of pregnancy.

**Table 2 microorganisms-10-00176-t002:** Frequency of distribution (%) of IgG antibodies to HSV-1 among pregnant women in the second trimester of pregnancy.

Groups	Titers of IgG Antibodies									
	High				Moderate				Low	
	1:12,800		1:6400		1:3200		1:1600		1:800	
	Abs	%	Abs	%	Abs	%	Abs	%	Abs	%
HSV-1 recurrence	4	4.4	5	5.6 *	8	8.8 *	32	35.6 *	41	45.6
**Total**			9	10 *			40	44.4	41	45.6
No recurrence of HSV-1			10	33.3	17	56.7	3	10		
**Total**			10	33.3			20	66.7		

Note: * *p* < 0.05 significance of differences between pregnant women with recurrent HSV-1 and in the group with no recurrence of HSV-1 in the first trimester of pregnancy.

**Table 3 microorganisms-10-00176-t003:** Frequency of distribution (%) of IgG antibodies to HSV-1 among pregnant women in the third trimester of pregnancy.

Groups	Titers of IgG Antibodies									
	High				Moderate				Low	
	1:12,800		1:6400		1:3200		1:1600		1:800	
	Abs	%	Abs	%	Abs	%	Abs	%	Abs	%
HSV-1 recurrence	3	3.3	21	23.3	33	36.7 *	27	30 *	6	6.7
**Total**			24	26.6			60	66.7	6	6.7
No recurrence of HSV-1			12	40	15	50	2	6.7	1	3.3
**Total**			12	40			17	56.7	1	3.3

Note: * *p* < 0.05 significance of differences between pregnant women with recurrent HSV-1 and in the group with no recurrence of HSV-1 in the first trimester of pregnancy.

**Table 4 microorganisms-10-00176-t004:** Serological receptivity to HSV-1 infection in women at different stages of pregnancy.

Groups	Titers of IgG Antibodies					
	High		Moderate		Low	
	RR	95% CI	RR	95% CI	RR	95% CI
HSV-1 recurrence						
First trimester	1.9 *	1.48–2.45				
Second trimester					1.67 **** 2.37 *****	1.27–2.21 1.84–3.03
Third trimester			1.67 ** 1.6 ***	1.21–2.32 1.16–1.22		
No recurrence of HSV-1						
First trimester			2.14 ^###^	1–4.7	1.69 ^####^	1.01–2.85
Second trimester	2.23 ^#^	1.52–3.28				
Third trimester	2.41 ^##^	1.62–3.58				

Note: * comparison of high titers of IgG antibodies between the first and second trimesters in the group with recurrent HSV-1; ** comparison of moderate titers of IgG antibodies between the third and first trimesters in the group with recurrent HSV-1; *** comparison of moderate titers of IgG antibodies between the third and second trimesters in the group with recurrent HSV-1; **** comparison of low titers of IgG antibodies between the second and first trimesters in the group with recurrent HSV-1; ***** comparison of low IgG antibody titers between the second and third trimesters in the HSV-1 recurrent group. ^#^ comparison of high titers of IgG antibodies between the first and second trimesters in the group without recurrent HSV-1; ^##^ comparison of high titers of IgG antibodies between the first and third trimesters in the group without recurrence of HSV-1; ^###^ comparison of moderate IgG antibody titers between first and third trimesters in the group without recurrence of HSV-1; ^####^ comparison of low IgG antibody titers between first and third trimesters in the group without recurrence of HSV-1.

**Table 5 microorganisms-10-00176-t005:** Socio-demographic characteristics and pregnancy parity in the study groups.

	HSV-1					
Variables	Recurrence		No Recurrence		Seronegative	
	Abs	%	Abs	%	Abs	%
Social status						
Employees	75	83.3	27	90	26	86.7
Housewives	11	12.2	2	6.7	2	6.7
Students	4	4.4	1	3.3	2	6.7
Demographic status: city residents	90	100	30	100	30	100
Maternal hereditary condition						
Hypertensive disease	21	23.3	6	20	5	16.7
Oncopathology	11	12.2	2	6.7	3	10
Diabetes mellitus	6	6.7	3	10	4	13.3
Congenital heart disease	3	3.3	1	3.3	-	-
**Total**	39	43.3	12	40	12	40
Father’s hereditary condition						
Hypertensive disease	7	7.8	3	10	3	10
Oncopathology	5	5.6	1	3.3	1	3.3
Diabetes mellitus	2	2.2			1	3.3
Congenital kidney disease	1	1.1				
**Total**	15	16.7	4	13.3	5	16.7
Planned pregnancy	9	10 *^#^	21	70	22	73.3
Primary pregnant	36	40	17	56.7	16	53.3
Primiparous	29	32	10	33.3	9	30
Multiparous	25	27.8 *^#^	3	10	5	16.7
Pregnancy parity						
Spontaneous miscarriage	20	22.2 *^#^	3	10	1	3.3
Missed abortion	4	4.4				
Legal abortion	34	37.8	13	43.3	12	40
Ectopic pregnancy	4	4.4	1	3.3		
Premature birth	3	10				
**Total**	65	78.8 *^#^	17	56.6	13	43.3

Note: * *p* < 0.05 differences in the group with recurrent HSV-1 during pregnancy compared with the group of seronegative pregnant women; ^#^ *p* < 0.05 compared to the group with no recurrence of HSV-1 during pregnancy.

**Table 6 microorganisms-10-00176-t006:** The frequency and structure of somatic and gynecological diseases in the study groups.

	HSV-1					
Variables	Recurrence		No Recurrence		Seronegative	
	Abs	%	Abs	%	Abs	%
Chronic pyelonephritis N11.1	21	23.3 ^#^	1	3.3		
Chronic bronchitis J42	12	13.3	3	10		
Chronic sinusitis J32.9	5	5.6	1	3.3		
Chronic tonsillitis J35.0	10	11.1			1	3.3
Chronic pharyngitis J31.2	4	4.4	1	3.3	1	3.3
**Total**	52	57.8 *^#^	6	20	2	6.7
ARI J00	59	65.6 *^#^	9	30	4	13.3
Acute sinusitis J 01	7	7.8				
Acute bronchitis J 20	3	3.3	1	3.3		
Exacerbation of chronic bronchitis	2	2.2				
**Total**	71	79 *^#^	12	33.3	4	13.3
Cervical erosion and ectropion N86	26	29 *	4	13.3	2	6.7
Chronic salpingo-oophoritis N70.1	19	21.1 ^#^	1	3.3		
Acute vaginitis N76.0	8	9				
Chronic inflammatory disease of the uterus N71.1	6	6.7	2	6.7		
Oligomenorrhea N91	5	5.6	2	6.7	1	3.3
**Total**	64	71.1 *^#^	9	30	3	10

Note: * *p* < 0.05 differences in the group with recurrent HSV-1 during pregnancy compared with the group of seronegative pregnant women; ^#^ *p* < 0.05 compared to the group with no recurrence of HSV-1 during pregnancy.

**Table 7 microorganisms-10-00176-t007:** The frequency and structure of pregnancy complications in the study groups.

	HSV-1					
Variables	Recurrence		No Recurrence		Seronegative	
	Abs	%	Abs	%	Abs	%
Threatened abortion O20.0	46	51.1 ^#^	10	33.3	8	26.7
Premature birth without delivery O60.0	19	21.1 *^#^	1	3.3	1	3.3
Anemia complicating pregnancy, childbirth and the puerperium O99.0	66	73.3 *^#^	6	20	4	13.3
Chronic placental insufficiency	65	72.2 *^#^	3	10	1	3.3
compensated form	55	84 *^#^	5	16.6	4	13.3
subcompensated form	9	13.8	2	6.7		
decompensated form	1	1.1				
Chronic intrauterine fetal hypoxia	65	72.2 *^#^	3	10		
Insufficient growth of the fetus, requiring the provision of medical care to the mother O36.5	7	7.7	1	3.3		
**Total**	333	297.6 *^#^	24	86.6	14	46.7

Note: * *p* < 0.05 differences in the group with recurrent HSV-1 during pregnancy compared with the group of seronegative pregnant women; ^#^ *p* < 0.05 compared to the group with no recurrence of HSV-1 during pregnancy.

**Table 8 microorganisms-10-00176-t008:** The frequency and structure of complications in childbirth in the study groups.

	HSV-1					
Variables	Recurrence		No Recurrence		Seronegative	
	Abs	%	Abs	%	Abs	%
Complications						
Abnormalities of forces of labor O62	22	28 ^#^	3	4.5	1	3.7
Premature rupture of membranes O42	11	14	1	3.8	1	3.7
Postpartum hemorrhage O72	4	5.1				
Placenta accreta O43.2	2	7.7				
Chronic inflammatory disease of the uterus N71.1	17	19	3	11.5		
**Total**	37	46.8 *^#^	4	15.4	2	7.4

Note: * *p* < 0.05 differences in the group with recurrent HSV-1 during pregnancy compared with the group of seronegative pregnant women; ^#^ *p* < 0.05 compared to the group with no recurrence of HSV-1 during pregnancy.

**Table 9 microorganisms-10-00176-t009:** Apgar score and weight distribution of newborns in the study groups.

	HSV-1					
Variables	Recurrence		No Recurrence		Seronegative	
	Abs	%	Abs	%	Abs	%
Apgar score at 1 min (points)						
10–7	76	85.3	28	93.3	30	100
4–7	14	15.4	2	6.7		
3 and below	1	1.1				
Apgar score at 5 min						
10–7	90	99	29	96.7	30	100
6–4	1	1.1	1	3.3		
3 and below						
Body weight (g)						
2000 and less	2	2.2				
2000–2499	3	3.3				
2500–2999	21	23.1	3	10	2	6.7
3000–3499	34	37.8	13	43.3	14	46.7
3500–3999	21	23.3	12	40	9	30
4000 and more	10	11.1	2	6.7	5	16.7

**Table 10 microorganisms-10-00176-t010:** The frequency and structure of diseases of newborns of the study groups.

	HSV-1					
Variables	Recurrence		No Recurrence		Seronegative	
	Abs	%	Abs	%	Abs	%
Cerebral ischemia P91.0	24	2.4 *^#^	2	6.7	1	3.3
Respiratory distress syndrome of the newborn P22.0	18	19.8 *^#^	1	3.3	1	3.3
Neonatal jaundice P59.2	6	6.6				
**Total**	48	52.8 *^#^	3	10	2	6.7
Perinatal infections						
Skin vesicles (vesiculosis)	19	21	2	6.7		
Pyoderma	8	8.8				
Neonatal conjunctivitis	6	6.6	1	3.3		
Omphalitis	1	1.1				
Sepsis	1	1.1				
Meningitis	1	1.1				
Pneumonia	1	1.1				
**Total**	37	40.6 ^#^	3	10		

Note: * *p* < 0.05 differences in the group with recurrent HSV-1 during pregnancy compared with the group of seronegative pregnant women; ^#^ *p* < 0.05 compared to the group with no recurrence of HSV-1 during pregnancy.

**Table 11 microorganisms-10-00176-t011:** Risk factors for the development of complications of pregnancy, childbirth and the early postpartum period, depending on the titers of IgG antibodies in women with recurrent HSV-1 during pregnancy.

Variables	Titers of IgG Antibodies					
	High		Moderate		Low	
	RR	95% CI	RR	95% CI	RR	95% CI
First trimester						
Threatened abortion O20.0					1.85	1.40–2.45
Anemia complicating pregnancy, childbirth and the puerperium O99.0	2.25	1.56–3.24	2.10	1.40–2.89	3.07	2.14–4.24
Chronic placental insufficiency	2.18	1.52–3.11	1.94	1.36–2.77	2.97	2.08–4.24
Chronic intrauterine fetal hypoxia					1.64	1.25–2.16
Second trimester						
Threatened abortion O20.0	2.38	1.82–3.10				
Premature birth without delivery O60.0	1.45	1.07–1.97				
Anemia complicating pregnancy, childbirth and the puerperium O99.0	3.85	2.68–5.53	1.92	1.34–2.75	1.88	1.31–2.69
Chronic placental insufficiency	3.72	2.62–5.30	1.86	1.30–2.65	1.82	1.28–2.58
Chronic intrauterine fetal hypoxia	1.9	1.52–2.42				
Abnormalities of forces of labor O62	1.56	1.17–2.07				
Third trimester						
Premature birth without delivery O60.0					1.66	1.26–2.19
Anemia complicating pregnancy, childbirth and the puerperium O99.0	2.75	1.91–3.96	2.01	1.40–2.89	4.13	2.88–5.91
Chronic placental insufficiency	2.66	1.86–3.80	1.94	1.36–2.77	3.99	2.81–5.67
Chronic intrauterine fetal hypoxia	1.45	1.09–1.92			2.33	1.82–1.98
Chronic inflammatory disease of the uterus N71.1					1.59	1.18–2.14

**Table 12 microorganisms-10-00176-t012:** Risk factors for the development of complications of the early antenatal period, depending on the titers of IgG antibodies in women with recurrent HSV-1 during pregnancy.

Variables	Titers of IgG Antibodies					
	High		Moderate		Low	
	RR	95% CI	RR	95% CI	RR	95% CI
Second trimester						
Cerebral ischemia P91.0	1.45	1.07–1.97			1.82	1.41–2.34
Respiratory distress syndrome of the newborn P22.0	1.45	1.07–1.97			1.63	1.22–2.16
Perinatal infections						
Skin vesicles (vesiculosis)	1.45	1.07–1.97				
Third trimester						
Perinatal infections						
Skin vesicles (vesiculosis)					1.67	1.26–2.19

## Data Availability

Not applicable.

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
