# Peer review of "The Effect of HSV-1 Seropositivity on the Course of Pregnancy, Childbirth and the Condition of Newborns"

_microorganisms, 2022, doi:10.3390/microorganisms10010176_

Round 1
Reviewer 1 Report
The serological test is now described in sufficient detail.
The concept of latency is not presented as it is common in the textbook literature. Eg. on lines 190-191 and in many other occasions, recurrent infection is presented as an opposite to latent infection., However, it is the latent infection that activates in recurrences and the reactivation does not cure the latent infection, ie., those with recurrent herpes do harbor latent HSV and will have it even after the recurrences. In other words, all the HSV seropositive individuals harbor the latent virus, whether they reactivate it often or not. The forms of infection should be designated differently, not as alternative forms of infection.
Author Response
См вложение.

Reviewer 2 Report
The updated version of the manuscript is much improved in terms of formatting and readability. My only major suggestion would be to again ensure that the comma and period format for decimals is consistent, in particular Table 7 is with commas while the rest are periods.
Reviewer 3 Report
The study entitled “The effect of HSV-1 seropositivity on the course of pregnancy, childbirth and the condition of newborns” by Dr. Andrievskaya and collaborators analyzed the presence and serum levels of IgG antibodies against HSV-1 in two groups of pregnant females, the longitudinal evaluation of HSV1 has also bee performed during rpegnancy. While the ms might reach interest across the readers as reporting very interesting findings regarding the anti-HSV-1 immunological state during pregnancy, the ms is in general poorly written and several sections should be reorganized. Several sections present out of place revision that should be removed, while sections reorganized
I therefore recommend a major reorganization
General comments
• Methods, additional information on females should be included, such as main age, pregnancy outcome etc
• “Ig G” should be quoted as “IgG”. For instance, PMID: 34961780. Please revise the entire text, accordingly
• A comparison between serological data form first and first/second/third trimester should be included in the result section. A figure depicting this longitudinal status would be helpful for the reader. The longitudinal status of pregnant females might be an interesting indicator of HSV-1 infection susceptibility during the different phases of gestation
• Lines 114-158 and 177-186, as well as others sections throughout the text, it is clear that revision marks are still present, please fix this problem
• Results reporting the CIs are quite often difficult to read and they should be shortened/removed. Including CIs only in tables can improve the reading of the results. This reviewer also suggests removing p values and CIs from the discussion, otherwise it seems a repetition of the result section.
• In the discussion section, the results should be more deeply discussed and compared with those already reported in the literature
Minors
Lines 17-21 reading the results reported in the abstract it is unclear wither low or high IgG levels are linked to the physio pathological conditions listed. I suggest reformulating the sentence
Lines 38-30 the immune modulation during pregnancy might be accountable for an increase in susceptibility for viral infections during pregnancy, as described in this recent immunological study (doi.org/10.3389/fmicb.2021.789991). This notion should be included
Lines 64-36 please include supporting reference
Lines 46-49 the prevalence of circulating anti-HSV antibodies? In other words, are those reported seroprevalences?
Lines 98-99 “Serological status……120 pregnant women seropositive to HSV-1 and 30 pregnant women seronegative to HSV-1 and HSV-2 pregnant women.” This sentence should be refreshed being unclear. In addition, how the positivity condition has been assessed? With the kits described in the methods? If yes, this is a result and the sentence should be modified as, for instance, “Serological status….in 150 pregnant women”.
Line 544 this might be related to different populations included which reflect different geographical regions, or different assays for detecting HSV serology. Please improve this point
Line 561-566 this result should be discussed and compared to the literature. The same coment can be expanded for the entire discussion in general
Author Response
Please see the attachement.

Round 2
Reviewer 3 Report
Following revision, the author's have addressed the concerns by the reviewers. The ms can be accepted in the present form.
This manuscript is a resubmission of an earlier submission. The following is a list of the peer review reports and author responses from that submission.
Round 1
Reviewer 1 Report
Authors showed effect of HSV-1 seropositivity on the course of pregnancy, child birth, and the conditions of newborns. Low or high titers of IgG antibodies to HSV-1 determined by the timing of infection recurrence during pregnancy are associated with a high incidence of somatic pathology, complications of pregnancy, childbirth, and the neonate period.
This paper is too complicated to understand properly. It should be much simplified.
In the statistical analysis, there are some problems. Authors have used Student’s t-test to analyze three groups (HSV-1 recurrence, latent, and seronegative). Authors also used Pearson's chi-square test. But this test is not suitable for small sample.
Authors used IgG antibodies to HSV-1. Antibodies to HSV-1 usually cross-react to those to HSV-2. Please indicate the method of measurement, and show that your method has no cross-reactivity to HSV-2 antibodies.
Recurrent infections during pregnancy were diagnosed with the presence of type-specific IgM antibodies to HSV-1. The presence of type-specific IgM antibodies to HSV-1 usually indicates not recurrent infection but initial infection of HSV-1.
Table 2, Table 3, Table 6, and Table 8 have many incorrect data (especially number, *, **, ***).
Line 156: first trimester?? Second trimester?
Line 259: ARI were detected in 79%?? 65.6%?
Author Response
Просмотрите приложение.

Reviewer 2 Report
In this paper, the authors use modern cohorts of pregnant women to track the outcomes of recurrent and latent HSV-1 infections on birth. Overall, they found that low IgG titers correlated with increased a range of birth complications and neonatal disease. The study groups are reasonably matched and the analytical methods are appropriate while the findings are of relevant clinical interest.
The data discussed in section 3.5 regarding antibody titers and childbirth outcomes is one of the major findings of the paper and should be included in a table.
There are otherwise some minor formatting issues:
Please check the consistency for decimal formats. Several paragraphs interchange "." with "," for percentages. It would perhaps be best to choose the period "." for the text and tables.
Insert a space in IgG on line 17 for consistency
The sentence beginning on line 35 with "12-25%" should specify infected adults or total adults
Line 38 may be easier to read by replacing the comma in ", with opportunistic" to "and with opportunistic"
Line 141, what exactly is meant by a "different type of immune reactivity?" Different from other pathogens?
Line 229, pluralize, or delete, "member"
The sentence of lines 231-232, change to "indicated hereditary burdens by the mother and father with the same frequency"
Line 434: please check the percentage for HSV-2, the upper bound for infection rate for a country as now it suggests that the entire population is positive specifically for HSV-2.
Line 437, change to "up to 30% HSV infections occur with HSV-1"
Reviewer 3 Report
The manuscript presents association of different clinical stages of HSV-1 infection with some clinical features of pregnancy and the condition of the newborns. There is some limited novelty in evaluation of herpes simplex virus type 1 serostatus in the context of pregnancy. However, there are methodological deficiencies and nonconventional use of the concept of latency of HSV. Everyone who experiences reactivating/recurring HSV infection does harbor the latent virus. Therefore, those reactivating often or less often, do all harbor the latent HSV, which is the source of reactivations. So just one group of study population can not be designated as "women with latent HSV infection". All seropositive women possessed the latent virus. The elementary method of the antibody testing was not described at all. Is the test commercial? Which is the antigen, how is it prepared? Is the test type-common or type-specific? Which is the protocol? This all should be described in the section 2: Materials and methods. The Results section presents, on the line 123, a statement on HSV type-specific IgM antibodies. It is generally known that IgM class antibodies are not HSV type specific. How is that test performed, how is the type specificity validated? It is well known that many HSV recurrences do not elicit detectable IgM antibody response. Therefore these results would be unexpected and nonconventional.
Round 2
Reviewer 1 Report
Previous reviewer’s question
Authors used IgG antibodies to HSV-1. Antibodies to HSV-1 usually cross-react to those to HSV-2. Please indicate the method of measurement, and show that your method has no cross-reactivity to HSV-2 antibodies.
Authors’ answers
Serological tests of HSV description of commercial ELISA kits. The determination of nonspecific to HSV-1, 2 and specific to HSV-2 IgM and G was carried out.
New reviewer’s comment
It is not sufficient to indicate that authors’ HSV antibody analyses were derived from only HSV-1. Please show the epitope of used antibodies. Please indicate the specificity of both nonspecific to HSV-1, 2 kit and HSV-2 specific kit. Reviewer thinks the specificity of HSV-2 kit might be very low. Were HSV-2 positive sample not detected at all in this study?
Authors’ answers
Four-fold or more increase in the titer of IgG antibodies to HSV-1, 2 in dynamics after 12 days and the absence of IgM and G to HSV-2 were treated as recurrence of HSV-1.
New reviewer’s comment
It is not clear whether four-fold or more increase in the titer of IgG antibodies to HSV-1, 2 indicates HSV recurrence correctly.